# LncRNA NEAT1 Silenced miR-133b Promotes Migration and Invasion of Breast Cancer Cells

**DOI:** 10.3390/ijms20153616

**Published:** 2019-07-24

**Authors:** Xinping Li, Siwei Deng, Xinyao Pang, Yixiao Song, Shiyu Luo, Liang Jin, Yi Pan

**Affiliations:** State Key Laboratory of Natural Medicines, Jiangsu Key Laboratory of Druggability of Biopharmaceuticals, School of Life Science and Technology, China Pharmaceutical University, 24 Tongjiaxiang Avenue, Nanjing 210004, China

**Keywords:** NEAT1, breast cancer, metastasis, miR-133b, translocase of inner mitochondrial membrane 17 homolog A (TIMM17A)

## Abstract

Breast cancer, the most prevalent cancer type among women worldwide, remains incurable once metastatic. Long noncoding RNA (lncRNA) and microRNA (miRNA) play important roles in breast cancer by regulating specific genes or proteins. In this study, we found miR-133b was silenced in breast cancer cell lines and in breast cancer tissues, which predicted poor prognosis in breast cancer patients. We also confirmed that lncRNA NEAT1 was up-regulated in breast cancer and inhibited the expression of miR-133b, and identified the mitochondrial protein translocase of inner mitochondrial membrane 17 homolog A (TIMM17A) that serves as the target of miR-133b. Both miR-133b knockdown and TIMM17A overexpression in breast cancer cells promoted cell migration and invasion both in vitro and in vivo. In summary, our findings reveal that miR-133b plays a critical role in breast cancer cell metastasis by targeting TIMM17A. These findings may provide new insights into novel molecular therapeutic targets for breast cancer.

## 1. Introduction

Breast cancer is the most frequently diagnosed cancer causing woman death. Thanks to improved diagnostic tools and treatment, breast cancer-caused deaths have decreased since the early 1990s. However, metastasis-preventing strategies are less effective, and metastasis is the leading cause of death amongst patients [1]. Therefore, reducing metastasis is urgent for improving breast cancer treatment outcomes.

Up to now, many molecular mechanisms and pathways of breast cancer metastasis have been investigated, but the role of non-coding RNA (ncRNA) involved in metastasis has not been fully elucidated. Among these ncRNAs, microRNAs (miRNAs), a group of noncoding small RNAs, and long non-coding RNAs (lncRNAs) have been drawn attention to [2,3,4]. Increasing evidence has uncovered the indispensable function of miRNAs in post-transcriptional regulation of oncogenes and tumor suppressor genes, thus modulating the biological behaviors of tumor cells such as invasion, metastasis, proliferation, and apoptosis [5,6,7]. LncRNAs are also known to have significant regulatory effects on carcinogenesis and cancer development by acting as scaffolds or guides to regulate interactions between proteins and genes, as decoys to bind proteins or miRNAs, and as enhancers to modulate transcription of their targets after being transcribed from enhancer regions or their neighboring loci [8,9]. Intriguingly, some recent studies reported that lncRNAs may not only bind miRNAs, but could also repress their expression by inducing miRNA degradation, leading to the up-regulation of their targets [10,11,12].

miR-133b, which participates in myoblast differentiation and myogenic-related diseases, is commonly recognized as a muscle-specific miRNA [13]. Recent reports demonstrated that miR-133b was dysregulated in many kinds of cancer and contributed to malignant progression via influencing cellular proliferation, apoptosis and motility [14,15]. However, the expression and function profile of miR-133b appears to be inconsistent among different cancers. For example, high miR-133b expression levels were found to be associated with poor prognosis for progression-free survival with bladder cancer, whereas its low expression levels in tumor tissues were found to be related to poor prognosis for overall survival and positive lymph node metastasis in colorectal cancer [16], and promote cell apoptosis, repress cell proliferation, tumor angiogenesis, drug-resistant and radioresistant, cell migration and invasion in lung cancer [17]. Despite these studies, whether miR-133b is involved in the development of breast cancer remains largely elusive.

In this study, we first validated the expression pattern and of miR-133b in breast cancer tissues and cells by using databases and qRT-PCR, and then we explored the regulatory mechanism of miR-133b silencing in breast cancer. Furthermore, we investigated the role of miR-133b in breast cancer cells by using various functional assays and revealed their underlying mechanism in vitro. Moreover, we confirmed miR-133b suppressed breast cancer metastasis through targeting TIMM17A in the mouse xenograft model. Our study firstly revealed that the NEAT1/miR-133b/TIMM17A axis was involved in breast cancer metastasis and might provide a potential target for breast cancer therapy.

## 2. Results

### 2.1. miR-133b is Down-Regulated in Breast Cancer Tissues and Cells

We first determined whether miR-133b has a specific expression profile in breast cancer tissues and cells, by searching StarBase public database. We found miR-133b levels were down-regulated in all the 14 types of common cancer tissues compared to their normal counterparts (Figure 1A), and remarkably lower in 1085 breast cancer tissues compared to 104 normal breast tissues according to the TCGA database (Figure 1B). We also evaluated the correlation between miR-133b expression and clinical outcomes from the TCGA database by Kaplan Meier Plotter (www.kmplot.com). As shown in Figure 1C, low expression of miR-133b predicted a poor prognosis in breast cancer patients (*n* = 916, *p* < 0.005). These data suggested miR-133b may act as a tumor suppressor gene in breast cancer.

We then validated that miR-133b is down-regulated in breast cancer. miR-133b levels were first determined in 40 paired breast cancer tissues and adjacent normal tissues by qRT-PCR. As shown in Figure 1D, miR-133b expression was significantly down-regulated in 92.5% (37 of 40 paired) of the breast cancer tissues. We further divided the samples into high (above the median, *n* = 20) and low (below the median, *n* = 20) miR-133b expression groups and explored the correlation between miR-133b expression and the clinicopathological factors of breast cancer patients. As shown in Appendix A and Figure 1E, the miR-133b level was positively correlated with tumor histological and lymph node metastasis by χ^2^ tests. ISH analysis of miR-133b also showed reduced expression of miR-133b in the breast cancer tissues compared to the adjacent normal tissues (Figure 1F). Meanwhile, we measured the expression of miR-133b in different breast cancer cell lines (MCF-7, SKBR-3, MDA-MB-468, BT-549 and MDA-MB-231) and non-tumorigenic breast epithelial cell line (MCF-10A), and found miR-133b levels were not only lower in breast cancer cells than normal breast cells, but even lower in the more aggressive breast cancer cell lines (MDA-MB-231, MDA-MB-468, BT-549 and SKBR-3) than the less aggressive one (MCF-7) (Figure 1G). Overall, the results suggested that the expression level of miR-133b is decreased in breast cancer tissues and cells and may serve as an independent predictor for overall survival in breast cancer and act as a tumor suppressor gene involved in breast cancer metastasis.

### 2.2. LncRNA NEAT1 Silences miR-133b Expression in Breast Cancer Cells

We then investigated the mechanism by which miR-133b expression is down-regulated in breast cancer cells and tissues. LncRNAs usually act as competing endogenous RNAs (ceRNA) by binding miRNAs and could even repress their expression [10]. Ago crosslinking-immunoprecipitation and high-throughput sequencing (CLIP-seq) data in the StarBase indicated that lncRNA nuclear paraspeckle assembly transcript 1 (NEAT1) might interact with miR-133b, and the predicted potential binding site between NEAT1 and miR-133b was illustrated in Figure 2A.

To validate whether NEAT1 is associated with the down-regulation of miR-133b in breast cancer cells, we first identified specific interactions between NEAT1 and miR-133b by luciferase reporter assay. Figure 2A showed that miR-133b overexpression reduced the luciferase activities of the NEAT1-WT reporter vector but not the NEAT1-MUT reporter. RIP assays performed on MCF-7 and MDA-MB-231 cell extracts using antibodies against Ago2 further demonstrated that NEAT1 and miR-133b were all enriched in Ago2-immunoprecipitation (Ago2-IP) relative to the control (IgG-IP; Figure 2B). Moreover, NEAT1 overexpression decreased the levels of miR-133b while NEAT1 knockdown increased the levels of miR-133b in MCF-7 and MDA-MB-231 cells (Appendix A and Figure 2C). These results suggested that NEAT1 might repress the expression of miR-133b via direct binding at the specific site.

In most cases, lncRNAs generally have expression patterns that are opposite to their target miRNAs [2,18]. Therefore, we then investigated whether NEAT1 expression is also inversely correlated with miR-133b expression in breast cancer cells and clinical specimens. By measuring the expression levels of NEAT1 in different breast cancer cell lines and normal breast cell line, we found NEAT1 levels were up-regulated in breast cancer cells compared with normal breast cells (Figure 2D) and negatively correlated with miR-133b levels (Figure 2E). Importantly, we found that NEAT1 was up-regulated in 90% (36 of 40 paired) of breast cancer tissues compared with adjacent normal tissues (Figure 2F), and NEAT1 levels were negatively correlated with miR-133b levels in breast cancer tissues (Figure 2G). By using Kaplan Meier Plotter, we evaluated the correlation between NEAT1 expression and clinical outcomes from the TCGA database. As shown in Figure 2H NEAT1 overexpression predicts a poor prognosis in breast cancer patients (*n*  = 2519, *p* < 0.005). These findings indicated that NEAT1 might act as an oncogene by down-regulating the expression of miR-133b.

### 2.3. Silencing of miR-133b Expression Promotes Breast Cancer Migration and Invasion

To identify the role of miR-133b in breast cancer progression, we introduced the inhibitor or mimics of miR-133b into MCF-7 and MDA-MB-231 cells. (Appendix A). The wound healing assays (Figure 3A,B) and transwell assays (Figure 3C,D) showed that the migration and invasion capacities of MCF-7 and MDA-MB-231 cells were significantly reduced by the overexpression of miR-133b, but strongly enhanced by depleting miR-133b. Our data demonstrated that down-regulation of miR-133b was involved in breast cancer metastasis.

NEAT1 was also reported to promoted breast cancer cell migration and invasion [19,20]. To confirm the upstream regulatory effect of NEAT1 on miR-133b, we performed a rescue assay with breast cancer cells by overexpressing NEAT1 or miR-133b and NEAT1 (Appendix A). The transwell migration and invasion assays showed that NEAT1 overexpression increased the migrated and invaded MCF-7 cell numbers, while ectopic expression of miR-133b attenuated NEAT1-mediated pro-migration and invasion in MCF-7 cells (Appendix A). Similarly, MDA-MB-231 cells transfected with miR-133b inhibitor and NEAT1 siRNA exhibited stronger migration and invasion abilities compared to cells transfected with NEAT1 siRNA. These results suggested that NEAT1 might contribute to the migration and invasion of breast cancer cells via suppressing miR-133b (Appendix A).

### 2.4. Identification of TIMM17A as a Direct Target Gene of miR-133b in Breast Cancer Cells

To investigate the underlying molecular mechanism by which miR-133b promotes breast cancer cell migration, invasion, proliferation and stemness, we performed the Venn diagram analysis of predicted miR-133b targets from four independent databases: TargetScan, miRanda, miRDB and PicTar. 111 mRNAs were found in the intersection part (Figure 4A). TIMM17A was selected for further experimental verification based on the following conditions (Appendix A): related to breast cancer, related to cancer migration or invasion, has not been reported as a miR-133b target, high expression in breast tumor (StarBase database), and correlated with poor survival of breast cancer patients (Kaplan Meier Plotter). TIMM17A was found to promote breast cancer tumorigenesis and metastasis [21], and its high expression is associated with adverse pathological and clinical outcomes in human breast cancer [22]. However, it has not been reported as the target of miR-133b. The predicted interaction between miR-133b and the target sites in the TIMM17A 3′-UTR was illustrated in Figure 4B. Perfect base-pairings between the seed region and the cognate target have been observed, and the free energy value of the hybrid was well within the range of genuine miRNA-target pairs (−19.5 kcal/mol). Subsequently, we confirmed that miR-133b directly targeted the predicted binding sites in the TIMM17A 3′-UTR by a luciferase reporter assay (Figure 4C). Moreover, we observed that TIMM17A mRNA and protein levels were significantly decreased in the cells transfected with the miR-133b mimics, while remarkably increased by suppression of the endogenous miR-133b (Figure 4E). The above results suggested that miR-133b down-regulated TIMM17A protein levels through directly binding to its 3′-UTR and degrading TIMM17A mRNA.

In most cases, miRNAs generally have expression patterns that are opposite to that of their targets. Therefore, we investigated whether the miR-133b expression is inversely correlated with TIMM17A expression in breast cancer cells and clinical specimens. We measured TIMM17A protein levels in the breast cancer cells and normal breast cells, and found TIMM17A protein levels were higher in the breast cancer cells (Figure 4F). By searching StarBase public database, we found TIMM17A mRNA levels were remarkably higher in 1104 breast cancer tissues compared to 113 normal breast tissues (Figure 4G), and its high expression predicts a poor prognosis in breast cancer patients (*n* = 2519, *p* < 0.005) using the Kaplan Meier Plotter (Figure 4H). By measuring the expression levels of TIMM17A in breast cancer tissues and adjacent normal tissues, we found TIMM17A levels were up-regulated in 87.5% (35 of 40 paired) of breast cancer tissues compared with adjacent normal tissues (Figure 4I) and negatively correlated with miR-133b levels (Figure 4J). Western blotting results showed TIMM17A protein levels were significantly increased in paired breast cancer tissues and adjacent normal tissues and TIMM17A protein levels were higher in all the breast cancer tissues than their adjacent normal tissues (Figure 4K). Taken together, these results indicated that TIMM17A is a direct target of miR-133b in breast cancer and might be correlated with poor outcome in breast cancer patients.

### 2.5. miR-133b Promotes Breast Cancer Cells Migration and Invasion Via Targeting TIMM17A

We next focused on studying whether silencing of miR-133b facilitates breast cancer cell migration and invasion by up-regulating TIMM17A expression. The rescue experiments were performed by transfecting breast cancer cells with miR-133b mimics or inhibitor, and TIMM17A vector or siRNA (Figure 5A). We observed that transfecting TIMM17A-overexpression plasmid markedly promoted MCF-7 and MDA-MB-231 cell migration and invasion, while transfection of TIMM17A siRNAs repressed it in MCF-7 cells (Figure 5B–E). Also, the cells transfected with miR-133b mimics and the TIMM17A-overexpression plasmid exhibited significantly higher migration and invasion abilities, while the cells transfected with miR-133b inhibitor and the TIMM17A siRNA showed significantly lower abilities than the cells transfected with the miR-133b inhibitor (Figure 5B–E). Collectively, these data demonstrated that miR-133b silencing promoted migration and invasion of breast cancer cells through targeting TIMM17A.

### 2.6. Targeting TIMM17A by miR-133b Promotes Breast Tumor Cell Metastasis in the Mouse Model

Finally, we investigated the role of miR-133b in mediating breast cancer cells metastasis in the mouse model by targeting TIMM17A (Figure 6A). We first generated five types of modified MDA-MB-231 cell lines, confirmed by qRT-PCR and western blotting (Figure 6B,C): cells infected with control lentivirus, cells stably transfected with miR-133b lentivirus, cells stably transfected with TIMM17A lentivirus, cells stably co-transfected with miR-133b and TIMM17A lentiviruses, and cells stably transfected with a miR-133b sponge. Subsequently, we injected these four modified MDA-MB-231 cell lines intravenously into female nude mice through the tail (Figure 6C). The metastasis was assessed by bioluminescent imaging (BLI) on day 5, 15 and 30 after implantation. As shown in Figure 6D and E, the imaging assay showed that the GFP-labeled migrated cells were mainly distributed in their lungs and livers. The fluorescent intensities in lung and liver were significantly weaker in the miR-133b-overexpressing MDA-MB-231 cell group and stronger in the TIMM17A-overexpressing or the miR-133b-knockdown MDA-MB-231 cell group compared to the control. Likewise, TIMM17A overexpression attenuated the inhibition of MDA-MB-231 cell metastasis caused by miR-133b-overexpression.

After 4 weeks, mice were killed and their whole lung tissues were harvested. The numbers of macroscopically visible tumor nodules on the lung surface were counted (Figure 6F), and the lung tissues were sectioned for H&E staining to evaluate tumor metastasis (Figure 6G). The results also showed significant differences in tumor number and growth features in these specimens (Figure 6F,G). In mice injected with TIMM17A-overexpressing or miR-133b-knocked down MDA-MB-231 cells, tumors with clear boundaries (arrows), higher quantities and size were found in the lungs, while only less and smaller tumor masses were scattered in the lungs of miR-133b over-expression group. Also, the numbers and size of tumors in the TIMM17A and miR-133b both over-expressed group were similar compared with the control group. These results supported the role of miR-133b in promoting breast cancer cell metastasis in mice through suppressing TIMM17A expression.

We also investigated the role of miR-133b/TIMM17A in mediating breast cancer cells metastasis in the mouse model by using four types of modified MCF-7 cell lines (Appendix A). The metastasis was assessed by BLI on day 15, 35 and 60 after implantation because of the low metastatic ability of MCF-7 cells. As shown in Appendix A, the fluorescent intensities in the lungs were significantly stronger in the miR-133b-knockdown MCF-7 cell group and weaker in the TIMM17A knockdown MDA-MB-231 cell group compared to the control. Likewise, TIMM17A knockdown attenuated the promotion of MCF-7 cell metastasis caused by miR-133b-knockdown. The numbers of macroscopically visible tumor nodules on the lung surface and the H&E staining results for the lung tissues displayed the same trend, which was consistent with the BLI assays. Taken together, these results supported the role of miR-133b in promoting breast cancer cell metastasis in mice through suppressing TIMM17A expression.

## 3. Discussion

In this study, we found that the expression of miR-133b was remarkably down-regulated in breast cancer tissues and cell lines, which was associated with poor prognosis in breast cancer patients. Also, we demonstrated that high expression of lncRNA NEAT1 suppress the expression of miR-133b in breast cancer. In vitro functional assays showed that miR-133b facilitated breast cancer cell migration and invasion. Mechanistically, we found that mitochondrial protein translocase of inner mitochondrial membrane 17 homolog A (TIMM17A) could be a target of miR-133b, and miR-133b silencing caused TIMM17A up-regulation leading to the metastasis of breast cancer both in vitro and in vivo. Our study has been summarized in a working model in Figure 6H.

Breast cancer can hardly be cured by traditional therapies. Therefore, researchers have focused their attention on therapy at the molecular level, such as finding potential biomarkers and gene therapies, which may be a new strategy to improve the therapeutical outcomes. miRNA microarray data and individual experiments have demonstrated that miR-133b was frequently down-regulated in various cancer types, such as colorectal cancer, bladder cancer, lung cancer, glioblastoma, ovarian cancer, prostate cancer, gastric cancer, head and neck cancer, renal cell carcinoma, hepatocellular carcinoma, and laryngeal cancer [23,24]. miR-133b mainly plays tumor suppressor roles in malignant tumors except for hepatocellular carcinoma and cervical carcinoma [25,26]. In this study, we consistently found the down-regulatory behavior of miR-133b in breast cancer, by exploring the TCGA database and detecting its expression levels in 40 breast cancer tissues and five breast cancer cell lines. Moreover, we observed that miR-133b had a low expression level in breast cancer patients with invasive ductal carcinoma or lymph node metastasis, and that low expression was associated with poor prognosis of patients with breast cancer. All of the above supported that miR-133b has a tumor-suppressing potential.

However, the regulatory mechanisms leading to abnormal expression of miR-133b have not been clearly stated. Histone methylation and acetylation might be associated with down-regulatory miR-133b in gastric cancer [27], and miR-133b dysregulation in colon cancer cells was allegedly due to the change of transcriptional activity caused by TAp63 [28]. Recent studies reported miR-133b-lncRNA ceRNAs regulatory network in colorectal cancer [29] and non-small cell lung cancer [30]. Until now, only a few articles reported the role of miR-133b in breast cancer [15]. Although it has been demonstrated that miR-133b is down-regulated in breast cancer and suppresses tumorigenesis and metastasis by targeting Sox9, the reason for miR-133b dysregulation has not been clearly illustrated. In this study, we proposed lncRNA NEAT1, which was up-regulated in breast cancer and also correlated with poor survival, could bind and even induce miR-133b degradation, thus down-regulated its expression in breast cancer cells. Meta-analyses and other studies showed that lncRNA NEAT1 is upregulated in various cancer entities resulting in an unfavorable prognosis as well as a poor overall survival [31]. A recent pan-cancer study predicted that NEAT1 is the lncRNA which has the most cancer gene targets [32]. Several studies demonstrated that NEAT1 contributed to breast cancer progression, and although it is a classical nuclear lncRNA, some of the studies have shown a connection between NEAT1 and different miRNAs, such as miR-218 [20], miR-448 [19], miR-101 [33] and miR-548 [34]. Consistent with our study, the expression levels of some abnormally expressed miRNAs in breast cancer were up- or down-regulated in response to NEAT1 knockdown or overexpression [20,33,35]. Also, NEAT1 was reported to promote cervical cancer development by binding and negatively modulating miR-133a expression, which belongs to the miR-133 family (including miR-133a and miR-133b) [36]. Besides, overexpression of lncRNA could decrease the copy numbers of its binding miRNA has been reported elsewhere [2,10,11,18]. However, the underlying mechanism is still elusive.Combing with our findings, it is suggested that NEAT1-mediated miRNA down-regulation might be a common mechanism for abnormal miRNA expression in human cancers.

Consistent with the previous study [15], in vitro research also showed significantly suppressed migration and invasion of breast cancer cells after up-regulating miR-133b. Mitochondrial protein TIMM17A contributes to a pre-protein import complex, which is essential for mitochondrial function, and mitochondrial dysfunction is considered as a hallmark of tumor pathogenesis [37]. However, TIMM17A has been rarely studied in cancers, and only two studies suggested that high TIMM17A expression is associated with adverse pathological and clinical outcomes in human breast cancer [22], and facilitates migration and invasion of breast cancer cells [21]. In this study, we proposed that TIMM17A could serve as a target of miR-133b, and it was required for breast cancer cell migration and invasion both in vitro and in vivo. Our data also revealed that TIMM17A was up-regulated in breast cancer, and its high expression predicted poor clinical outcome for breast cancer patients.

In conclusion, NEAT1-caused down-regulation of miR-133b promotes breast cancer metastasis through up-regulating TIMM17A, a target of miR-133b. Each of the three players in NEAT1/miR-133b/TIMM17A axis may become a novel therapeutic target for the treatment of breast cancer, which is of crucial significance for the clinical prevention and diagnosis of breast cancer.

## 4. Materials and Methods

### 4.1. Cell lines and Culture

The human breast cancer cell lines, MCF-7, MDA-MB-468 and SKBR-3, and the human embryonic kidney cell line HEK-293T were cultured in DMEM medium (Gibco, Thermo Fisher, Waltham, MA, USA) supplemented with 10% fetal bovine serum (FBS; Gibco, Thermo Fisher, Waltham, MA, USA). The human breast cancer cell line BT-549 was cultured in RPMI-1640 medium (Gibco, Thermo Fisher, Waltham, MA, USA), and the non-tumorigenic breast epithelial cell line MCF-10A was cultured in DMEM/Ham’s F-12 (1:1; Gibco, Thermo Fisher, Waltham, MA, USA) supplemented with 5% FBS. The human breast cancer cell line MDA-MB-231 was cultured in L-15 medium (Gibco, Thermo Fisher, Waltham, MA, USA) supplemented with 10% FBS. All the cells were purchased from the Institute of Biochemistry and Cell Biology of the Chinese Academy of Sciences (Shanghai, China), incubated at 37 °C with 5% CO_2_.

### 4.2. Patients and Clinical Specimens

All patient samples were collected from the Breast Disease Center of Jiangsu Province, First Affiliated Hospital of Nanjing Medical University (Nanjing, China) with written informed consent. The ethical approval was granted from Committees for Ethical Review in China Pharmaceutical University (Nanjing, China). Pathological diagnosis was made according to the histology of tumor specimens or biopsy and examined by experienced pathologists, and the clinicopathological characteristics are shown in the Appendix A. Breast cancer tissues and adjacent normal tissues were stored in liquid nitrogen. The study is compliant with all relevant ethical regulations for human research participants, and all the participants were provided with written informed consent.

### 4.3. RNA Extraction and Quantitative RT-PCR

Total RNA was isolated using TRIzol (Invitrogen, Carlsbad, CA, USA) and an RNeasy kit (QIAGEN, Hilden, Germany) with DNase I digestion according to the manufacturers’ instructions. Reverse transcription reaction was performed using PrimeScriptTM RT reagent Kit (Takara, Kusatsu, Shiga Prefecture, Japan), and diluted complementary DNA (cDNA) was used for qRT-PCR analysis using SYBR Premix Ex Taq II Kit (Takara, Kusatsu, Shiga Prefecture, Japan) with the appropriate primers listed in the Appendix A. For microRNA, stem-loop reverse transcription reactions were performed, and the 2^−∆∆Ct^ method was used to calculate the relative abundance of RNA genes compared with glyceraldehyde 3-phosphate dehydrogenase (GAPDH) or U6 expression.

### 4.4. Cell Transfection and Virus Infection

miRNA mimics, inhibitors, siRNAs (Genepharma, Shanghai, China) and plasmids transfections were performed using Lipofectamine 2000 (Invitrogen, Carlsbad, CA, USA) according to the manufacturer’s protocol. The miR-133b mimics, miR-133b inhibitor, NEAT1 siRNA (si-NEAT1), TIMM17A siRNA (si-MAML) and their negative controls were purchased from Shanghai GenePharma Co., Ltd. (Shanghai, China). The sequences of miR-133b mimics, miR-133b inhibitor, si-TIMM17A and their negative controls were list in the Appendix A. Lentivirus (Genepharma, Shanghai, China) encoding miR-133b or TIMM17A were imported into MDA-MB-231 cells as previously described [38]. The clones with the stable miR-133b or TIMM17A expression were selected by green fluorescence protein (GFP) expression.

### 4.5. Migration and Invasion Assays

To evaluate the migration and invasion ability of cells in vitro, wound healing assay, transwell migration assay and transwell invasion assay were performed based on published methods [38]. For the cell transmembrane migration assay, all the steps were carried out similarly to those in the invasion assay except for the Matrigel (BD Biosciences, Franklin Lakes, NJ, USA) coating. After incubation at 37 °C for 24 h or 48 h, the filters were removed. The cells adhering to the lower surface were fixed and stained with 0.1% Crystal Violet (Beyotime, Wuhan, China). To image the invaded or migrated cells, five randomly selected fields in each well were photographed and counted under an inverted microscope (Olympus, Wuhan, China) at the magnification of 200. All experiments were performed in triplicate.

### 4.6. Western Blotting

Cells were washed three times with cold phosphate buffered saline (PBS) and total cellular protein was extracted using a modified radioimmunoprecipitation assay (RIPA) lysis buffer on ice. 40 µg of proteins were separated with 10% sodium dodecyl sulfate-polyacrylamide gel electrophoresis (SDS-PAGE) gel and transferred onto a 0.22 mm polyvinylidene fluoride membrane (PVDF; Millipore, Burlington, MA, USA) using semi-dry transfer cells (Bio-Rad, Hercules, CA, USA). Antibodies used are listed in the Appendix A. The protein bands were visualized using the enhanced chemiluminescence (ECL) detection kit (Tanon, Shanghai, China). Finally, the proteins were normalized with GAPDH (1:1000; Santa) and visualized by ECL detection system (Tanon, Shanghai, China). The blots were analyzed using the ImageJ program (Wikimedia Foundation, San Francisco, CA, USA).

### 4.7. Luciferase Assay

The putative miR-133b binding sites of the human TIMM17A 3’UTR were synthesized and inserted between the SpeI and HindIII sites of the pMIR-Report plasmid (TIMM17A-WT-3′-UTR). We also constructed a pMIR-Report plasmid that carried the mutant TIMM17A 3′UTR region (TIMM17A-MUT-3′-UTR). For the luciferase reporter assay, HEK-293T cells were co-transfected with TIMM17A-WT-3′-UTR or TIMM17A-MUT-3′-UTR plasmids and mimics negative control (NC), miR-133b mimics, inhibitor NC or miR-133b inhibitor by using Lipofectamine 2000 (Invitrogen). The luciferase assay was performed 24 h after transfection using double-luciferase assay system (Beyotime, Wuhan, China).

### 4.8. RNA-Binding Protein Immunoprecipitation (RIP)

For Ago2-based RIP assays, MCF-7 and MDA-MB-231 cells were transfected with pcDNA3.1-TIMM17A or pcDNA3.1. After 48 h, cells were subjected to RNA-binding protein immunoprecipitation (RIP) experiments using an anti-Ago2 antibody and the Magna RIP™ RNA-Binding Protein Immunoprecipitation Kit (Millipore) according to the manufacturer’s instructions. qRT-PCR was performed to examine the expression levels of TIMM17A and miR-133b.

### 4.9. Xenograft Assays in Nude Mice

Female BALB/c nude mice (5–6 weeks, 18–20 g) were purchased from the Model Animal Research Center at Nanjing University (Nanjing, China) and kept under specific pathogen-free (SPF) conditions at China Pharmaceutical University. The procedures of all animal experiments complied with IACUC (Institutional Animal Care and Use Committee) regulations. For metastasis experiments, MDA-MB-231 cells were stably transfected with control lentivirus, miR-133b lentivirus, TIMM17A lentivirus, miR-133b plus TIMM17A lentivirus, and miR-133b sponge. The mice were randomly divided into four groups (five mice per group) and 0.1 mL PBS (containing more than 1 × 10^6^ cells) were injected intravenously into the tail. Metastases were then examined by bioluminescence imaging using an IVIS Spectrum Xenogen Imaging System (Xenogen, San Francisco, CA, USA) on day 10, 20 and 30. MCF-7 cells were stably transfected with control lentivirus, miR-133b sponge, TIMM17A sponge, and miR-133b plus TIMM17A sponge. The mice were randomly divided into four groups (five mice per group) and 0.1 mL PBS (containing more than 2 × 10^6^ cells) were injected intravenously into the tail. Metastases were then examined by bioluminescence imaging using an IVIS Spectrum Xenogen Imaging System (Xenogen, San Francisco, CA, USA) on day 15, 35 and 60. After scanning, intact lungs and livers were isolated from the mice and photographed. The tissues were excised and embedded in paraffin for histopathological examination. For all animal experiments, the operators and investigators were blinded to the group allocation. All animal experiments were approved by the Ethics Committee of China Pharmaceutical University (Permit Number: 2162326; Permit date: 1/27/2016).

### 4.10. Statistical Analysis

All data were shown as the mean and standard error of the mean (mean ± SEM, *n* = 3). Statistical analysis was performed using independent student *t*-test via GraphPad Prism 5.0. *p* < 0.05 was considered as statistically significant.

## Figures and Tables

**Figure 1 ijms-20-03616-f001:**
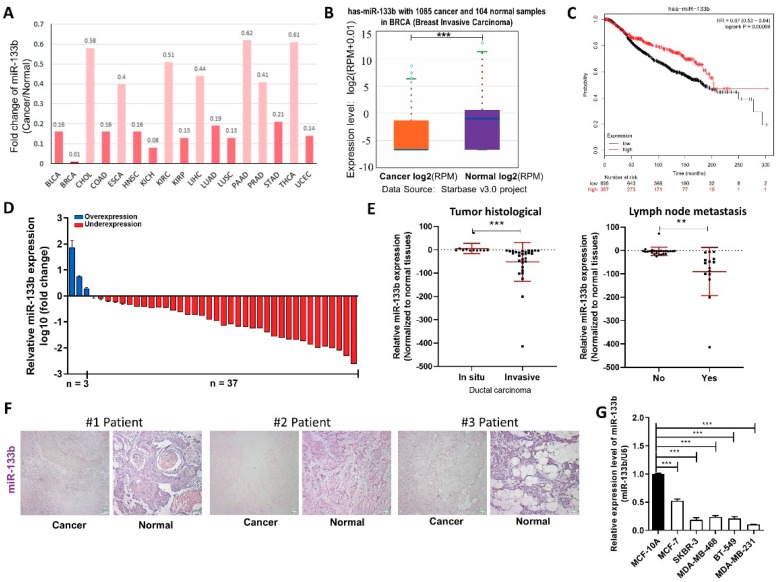
miR-133b is down-regulated in breast cancer cells. (**A**,**B**) Foldchange of miR-133b in 17 types of cancer tissues compared to their adjacent normal tissues (**A**) and expression levels of miR-133b in 1085 breast cancer tissues and 104 normal breast tissues (**B**) in StarBase public database from the TCGA project. (**C**) Overall survival analysis of breast cancer patients based on miR-133b expression (*n* = 916, log-rank test). Data was analyzed using Kaplan Meier Plotter (www.kmplot.com). (**D**) miR-133b levels were detected in 40 pairs of human breast cancer tissues and corresponding adjacent normal tissues by qRT-PCR. (**E**) miR-133b levels in breast cancer patients with different tumor histological (left), with (indicated with ‘yes’) or without (indicated with ‘no’) lymph node metastasis (right). (**F**) In situ hybridization (ISH) analysis of miR-133b expression levels in breast cancer tissues (Cancer) and their adjacent normal tissues (Normal). Representative locked nucleic acid *in situ* hybridization (LNA-ISH) images from patients #1, #2 and #3 are shown. Scale bar, 100 μm. (**G**) miR-133b levels were detected in different breast cancer cells compared with normal breast cells. * *p* < 0.05; ** *p* < 0.01; *** *p* < 0.001.

**Figure 2 ijms-20-03616-f002:**
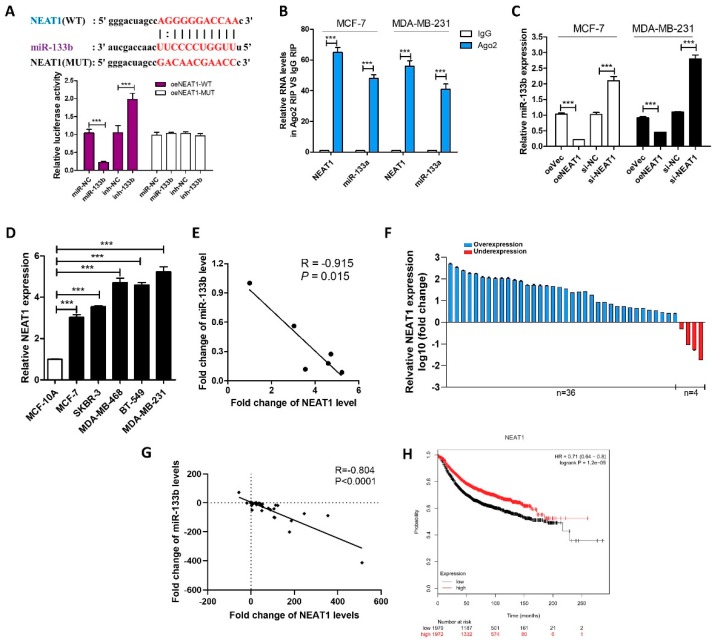
NEAT1 silences mir-133b expression in breast cancer cells. (**A**) Luciferase activity in MCF-7 cells co-transfected with a luciferase reporter containing NEAT1-WT or NEAT1-MUT (miR-133b-binding sequence mutated) and miR-133b. (**B**) Relative enrichment of NEAT1 and miR-133b associated with AGO2 in MCF-7 cells and MDA-MB-231 cells detected by anti-AGO2 RIP (non-specific IgG as negative control). (**C**) miR-133b levels in MCF-7 cells and MDA-MB-231 cells transfected with oeVec, oeNEAT1, si-NC or si-NEAT1. (**D**,**E**) NEAT1 levels (**D**) and Pearson’s correlation scatter plot of the fold change of NEAT1 and miR-133b levels (**E**) in different breast cancer cells and normal breast cells. (**F**) NEAT1 levels were detected in 40 pairs of human breast cancer tissues and corresponding adjacent normal tissues by qRT-PCR. (**G**) The correlation between miR-133b and NEAT1 in breast cancer tissues by Spearman correlation analysis. (**H**) Overall survival analysis of breast cancer patients based on NEAT1 expression (*n* = 3951, log-rank test). Data was analyzed using Kaplan Meier Plotter (www.kmplot.com). *** *p* < 0.001.

**Figure 3 ijms-20-03616-f003:**
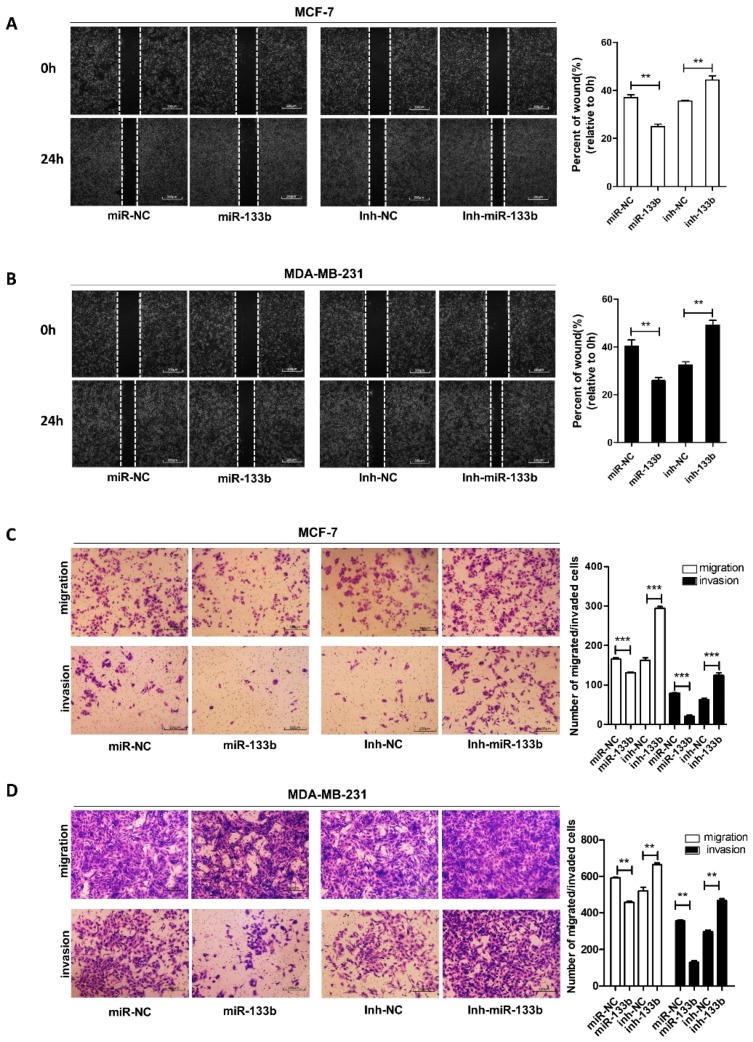
miR-133b promotes breast cancer cells migration and invasion *in vitro.* (**A**,**B**) Migration of MCF-7 cells (**A**) and MDA-MB-231 cells (**B**) transfected with mimics NC (miR-NC), miR-133b mimics (miR-133b), inhibitor NC (inh-NC) or miR-133b inhibitor (inh- miR-133b) detected by wound healing assay. Scale bar, 100 μm. (**C**,**D**) Migration and invasion of MCF-7 cells (**C**) and MDA-MB-231 cells (**D**) transfected with mimics NC, miR-133b mimics, inhibitor NC or miR-133b inhibitor detected by transwell migration assay and transwell invasion assay. Scale bar, 100 μm. ** *p* < 0.01; *** *p* < 0.001.

**Figure 4 ijms-20-03616-f004:**
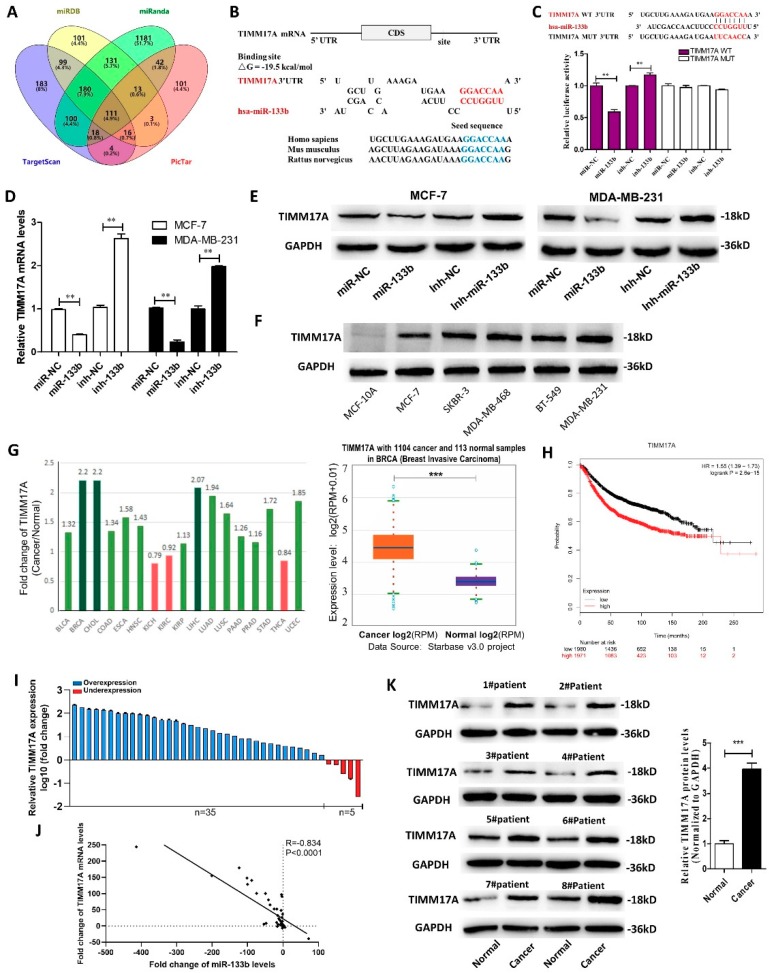
Identification of TIMM17A as a direct target gene of miR-133b in breast cancer cells. (**A**) Venn diagram analysis of four independent databases reveals 111 possible targets of miR-133b. (**B**) Schematic description of the hypothetical duplexes formed by the interactions between the binding sites in the TIMM17A 3′-UTR and miR-133b. The predicted free energy value of the hybrid is indicated. The seed recognition sites are denoted, and all nucleotides in these regions are highly conserved across species, including human, mouse, and rat. (**C**) Luciferase activity in MCF-7 cells co-transfected with a luciferase reporter containing either TIMM17A-WT or TIMM17A-MUT (miR-133b-binding sequence mutated) and mimics NC, miR-133b mimics, inhibitor NC or miR-133b inhibitor. Data are presented as the relative ratio of renilla luciferase activity and firefly luciferase activity. (**D**,**E**) TIMM17A mRNA (**D**) and protein (**E**) levels in MCF-7 cells and MDA-MB-231 cells transfected with mimics NC, miR-133b mimics, inhibitor NC or miR-133b inhibitor. (**F**) TIMM17A protein levels in different breast cancer cells and normal breast cells. (**G**) Left: Foldchange of miR-133b in 17 types of cancer tissues compared to their adjacent normal tissues. Right: miR-133b levels in 1104 breast cancer tissues and 113 normal breast tissues in starBase public database from the TCGA project. (**H**) Overall survival analysis of breast cancer patients based on TIMM17A expression (*n* = 3951, log-rank test). Data was analyzed using Kaplan Meier Plotter (www.kmplot.com). (**I**,**J**) TIMM17A mRNA levels (**I**) and Pearson’s correlation scatter plot of the fold change of TIMM17A mRNA 1 and miR-133b levels (**J**) in different breast cancer cells and normal breast cells. (**K**) TIMM17A protein levels in eight pairs of human breast cancer tissues (Cancer) and corresponding distal non-cancerous tissues (Normal). ** *p* < 0.01; *** *p* < 0.001.

**Figure 5 ijms-20-03616-f005:**
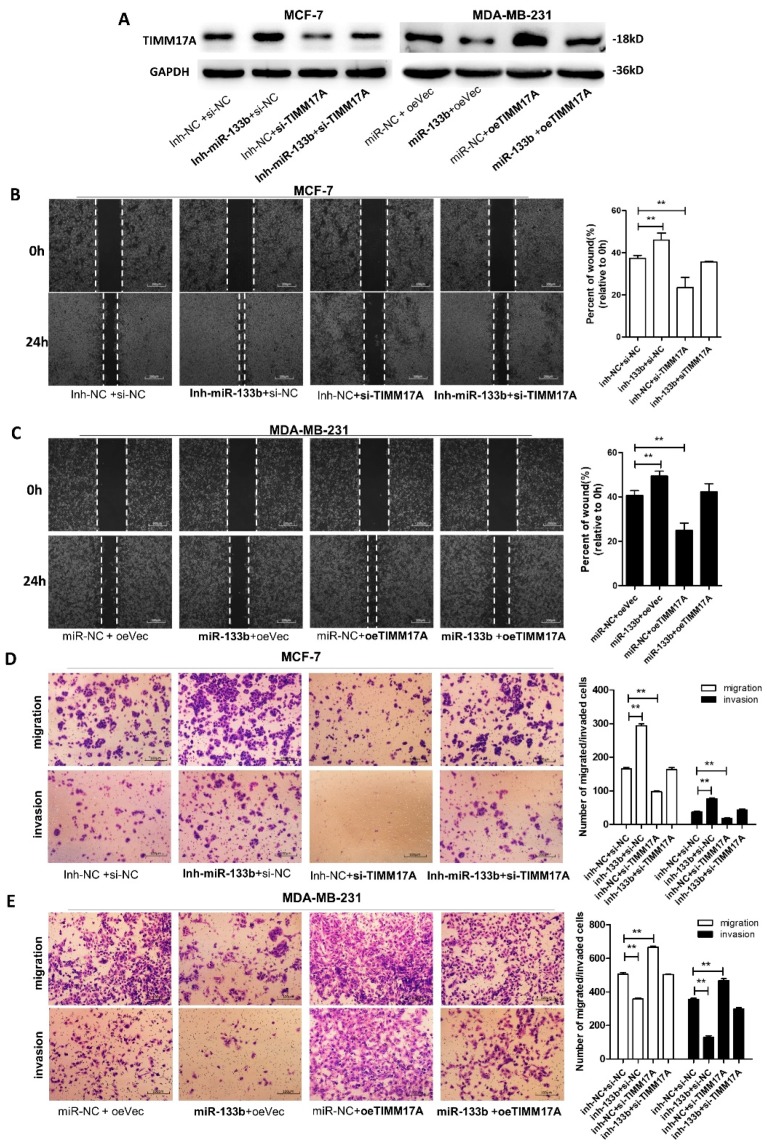
miR-133b promotes breast cancer cells migration and invasion via targeting TIMM17A (**A**) TIMM17A protein levels in MCF-7 cells transfected with inh-NC plus control siRNA(si-NC), inh- miR-133b plus si-NC, inh-NC plus TIMM17A siRNA (si-TIMM17A), or inh- miR-133b plus si-TIMM17A, and in MDA-MB-231 cells transfected with miR-NC plus control vector(oeVec), miR-133b plus oeVec, miR-NC plus TIMM17A vector (oeTIMM17A), or miR-133b plus oeTIMM17A. (**B**,**C**) Migration of MCF-7 cells transfected with inh-NC plus control siRNA(si-NC), inh- miR-133b plus si-NC, inh-NC plus TIMM17A siRNA (si-TIMM17A), or inh- miR-133b plus si-TIMM17A (**B**), and MDA-MB-231 cells transfected with miR-NC plus control vector(oeVec), miR-133b plus oeVec, miR-NC plus TIMM17A vector (oeTIMM17A), or miR-133b plus oeTIMM17A (**C**) detected by wound healing assay. Scale bar, 100 μm. (**D**,**E**) Migration and invasion of MCF-7 cells transfected with inh-NC plus control siRNA(si-NC), inh- miR-133b plus si-NC, inh-NC plus TIMM17A siRNA (si-TIMM17A), or inh- miR-133b plus si-TIMM17A (**D**), and MDA-MB-231 cells transfected with miR-NC plus control vector(oeVec), miR-133b plus oeVec, miR-NC plus TIMM17A vector (oeTIMM17A), or miR-133b plus oeTIMM17A (**E**) detected by transwell migration and invasion assay. Scale bar, 100 μm. ** *p* < 0.01; *** *p* < 0.001.

**Figure 6 ijms-20-03616-f006:**
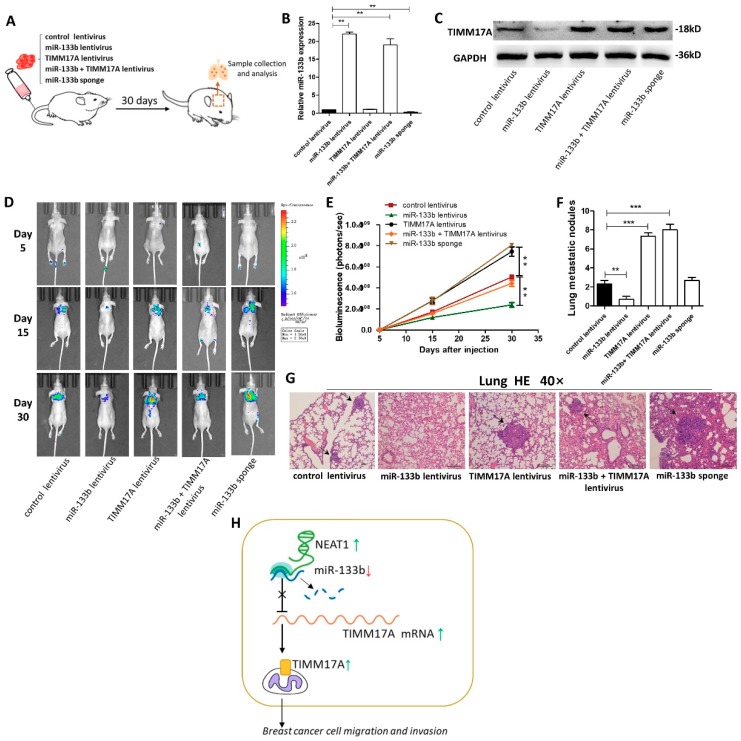
Effects of TIMM17A-targeted miR-133b on the lung and liver colonization of MDA-MB-231 cells xenografts in mice. (**A**) Experimental design: immunocompromised mice were injected through tail vein with MDA-MB-231 cells transfected with either the control lentivirus, miR-133b lentivirus, TIMM17A lentivirus, miR-133b lentivirus plus TIMM17A lentivirus, or miR-133b sponge. (**B**,**C**) miR-133b levels (**B**) and TIMM17A protein levels (**C**) in MDA-MB-231 cells transfected with either the control lentivirus, miR-133b lentivirus, TIMM17A lentivirus, miR-133b lentivirus plus TIMM17A lentivirus, or miR-133b sponge. (**D**,**E**) Representative BLI images (**D**) and quantitative analysis of the fluorescence intensities (**E**) of mice of five groups. The BLI was performed on days 5, 15, and 30 after injection. The intensity of BLI is represented by the color. (**F**,**G**) Numbers of metastatic nodules(**F**) and representative H&E-stained sections of lung tissues isolated from the intravenously injected mice. Black arrows indicate metastatic nodules. (**H**) A working model for the role of NEAT1-silenced miR-133b in breast cancer metastasis. During breast tumorigenesis, silencing of miR-133b mediated by NEAT1 facilitated migration and invasion of breast cancer cells through up-regulating the expression of TIMM17A. Scale bar, 200 μm. ** *p* < 0.01; *** *p* < 0.001.

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
