# Peer review of "LncRNA NEAT1 Silenced miR-133b Promotes Migration and Invasion of Breast Cancer Cells"

_ijms, 2019, doi:10.3390/ijms20153616_

Round 1

Reviewer 1 Report

In their manuscript the authors have reported on the role of miR-133b on metastasis in breast cancer through regulation of TIMM17A. The work reports some important findings and is thoroughly explained; however, i have some comments which are provided below:

·         Line 46: suggest saying “inconsistent among different cancers” instead of “various from different cancers”

·         In the introduction lines 45-50 the authors should also mention the role of miR-133b in lung cancer citation: Mol Ther Nucleic Acids. 2017 Sep 15; 8: 442–449

·         In line 333, the authors state that “Until now, only one article reported the role of miR‐133b in breast cancer[15].” I do not agree with this since there are several publications which report on the role of miR-133b in tumor suppression and metastasis control. These should all be included in the discussion section. You can find the citations below:

Oncol Rep. 2018 Feb; 39(2): 473–482

Clin Cancer Res. 2013 Aug 15;19(16):4477-87.

Author Response

We thank the reviewer for these constructive suggestions. We spent extensive work to finish the revision according to reviewer’s suggestions and revised the manuscript. The point-by-point response to the reviewer’s comments was in the attachment.

Reviewer 2 Report

Given the correlation of lower expression levels of miR-133b in cancer compared to healthy breast tissues, which is independent of the metastatic behavior, it would be interesting to assess the role of miR-133b in the regulation of cancer cell proliferation both in vitro and in vivo (proliferation assay and subcutaneous injection of breast cancer cell lines).

To confirm the upstream regulatory effect of NEAT1 on miR-133b, the authors should perform a rescue assay with cells overexpressing miR-133b and NEAT1.

In the results section 3.1 the sentence "Meanwhile, we measured the expression of miR‐133b in different breast cancer cell lines (MCF‐7, SKBR‐3, MDA‐MB‐468, BT‐549 and MDA‐MB‐231) and normal breast cell line (MCF‐10A), and found miR‐133b levels were not only lower in breast cancer cells than normal breast cells, but even lower in the more aggressive breast cancer cell lines (MDA‐MB‐231, MDA‐MB‐468, BT‐549 and SKBR‐3) than the less aggressive one (MCF‐7) (Fig. 1F).", should be referred to Fig. 1G.

In the results section 3.4 the sentence "By measuring the expression levels of NEAT1 in  87.5% (35 of 40 paired) of breast cancer tissues compared with adjacent normal tissues (Fig. 4I) and negatively correlated with miR‐133b levels (Fig. 4J)." should be referred TIMM17A expression. 

Fig. 6 should include also data performed on MCF7 breast cancer cells, to demonstrate that regulation of NEAT1/miR-133b/TIMM17A is able to reprogram non-metastatic cells to a metastatic phenotype.

From Fig. 6 it is clear that miR-133b is reducing the migratory potential of breast cancer cells, which are then able to grow in the lungs. It would be interesting to show the expression of miR-133b, or TIMM17A, in lung metastases to see which cells are able to form micro- and then macro-metastases, which is usually a feature of cells that are plastic in their epithelial-mesenchymal phenotype. Does miR-133b promote just a higher number of migratory cells or the induction of a partial E/M phenotype characterized by both proliferative and metastatic behavior?

The authors should give more emphasis to the therapeutic potential of the results here shown. In particular, in the "Discussion" section it should be indicated which of the 3 players here reported to regulate the metastatic dissemination f breast cancer cells, could reasonably be the target of therapy in the next future.

Minor English language revision and spell check are required.

Author Response

We thank the reviewer for these constructive suggestions. We spent extensive work to finish the revision according to reviewer's suggestions and revised the manuscript. The point-by-point response to the reviewer's comments was in the attachment.

Reviewer 3 Report

The paper by Li et al, discuss about the role of miR-133 b in breast cancer in the regulation of migration and invasion, through TIMM17A.

The paper has several major and minor issues that need to be solved to improve the overall meaning of the paper.

1-First of all, in line 3 fix the authors’ order, since it looks like the last author is missing.

2- The authors define as control MCF10A, which are defined as “normal”. I would add one of many available normal cell lines, while MCF10A is an  immortalized non transformed cell line.

3-Line 168, miR-133 b id DOWNREGULATED, while the authors say it’s up.

4-the cell lines chosen for the in vitro experiments are not the correct ones. The authors focused on MCF7 and MDA-MB-231 for both over expression and downregulation experiments. The correct model would be silencing miR-133b in MCF10A (or mcf7) and overexpress it in MDA-MD-231. Doing dowregulation experiments in mda-mb-231 makes no sense. This is true throughout the paper, and also the mice experiments should be fixed accordingly.  

5- NEAT1 being a ceRNA and acting downregulating miR-133b is an inference. Also, to be a ceRNA the two molecules should be positively correlated, not negatively. This sounds more like a target mechanism. Please clarify this point.

6- the choice of TIMM17A looks random, how did you pick the target out of the 111 available? (also, line 258, change miRNA in mRNA).

7- as regards the mice experiments, is not clear if you performed it also on MCF7 or not. They’re mentioned in the text, but figure 6 is only about mm231. Also, if they are, did you use the same timing? It is know that these cell lines have different metastatic abilities.

8- In the ISH (2F) you show a NEAT1 localization in the cytoplasm. NEAT 1 is a classical nuclear lncRNA. Is that a specific staining?

Author Response

(The authors gave the same response as above.)

Round 2

Reviewer 2 Report

Following the authors' adjustments, the manuscript has been improved both in term of quality and clarity for the readers. No other concerns from this reviewer.

Reviewer 3 Report

The authors provided detailed explanation in the revised version, and the modified manuscript is definitely improved.